# Analysis of Biogenic Secondary Pollution Materials from Sludge in Surface Waters

**DOI:** 10.3390/ijerph16234691

**Published:** 2019-11-25

**Authors:** Laima Česonienė, Edita Mažuolytė-Miškinė, Daiva Šileikienė, Kristina Lingytė, Edmundas Bartkevičius

**Affiliations:** Institute of Environment and Ecology, Faculty of Forest Science and Ecology, Vytautas Magnus University, Studentų str. 11, LT–53361 Akademija, Kaunas, Lithuaniadaiva.sileikiene@vdu.lt (D.Š.); Kristina.lingyte@vdu.lt (K.L.); edmundas.bartkevicius@vdu.lt (E.B.)

**Keywords:** lakes, sampling, nitrogen, phosphorus

## Abstract

Many countries of the world, including Lithuania, are making an effort to reduce surface water pollution. State monitoring data show that almost 80% of the lakes in Lithuania have an increased amount of sludge. One of the reasons for this increase in sludge is an excessive amount of biogenic material in the water. It is known that even after the source of pollution is removed, the condition of the lake water does not improve; rather, the condition of the lake water worsens due to the secondary pollution of sludge in the water. A study was conducted to determine the impact of secondary sludge pollution on water. For this study, 5 sludge samples were taken from different lakes in Lithuania. Fresh water was poured on the sludge samples, the concentrations of N_t_, NO_2_-N, NO_3_-N, NH_4_-N, PO_4_-P, P_t_, the pH and the changes in the electric conductivity (C) were measured in the water within 28 h. Research has shown that the thickness of the sludge layer influences the total amounts of nitrogen, phosphorus, and organic matter present in the sludge. As the thickness of the sludge layer increases in a lake, the total concentrations of nitrogen, total phosphorus and organic matter increase. Studies have also shown that the concentrations of all biogenic substances in water increase, with the exception of total phosphorus. This finding shows that organic phosphorus is "locked" in sludge, and no secondary pollution occurs from this source. Moreover, the electrical conductivity values of the water influence the release of biogenic substances from sludge in the water.

## 1. Introduction

Water quality depends mostly on the characteristics and quantities of pollutants entering a body of water. In Lithuania, as in many European countries, the main water polluters are industry, agriculture, and households [1]. The factors determining the quality of surface water can be divided into direct and indirect factors. Direct factors (rocks, soils, biota, and human activity) remove or supply soluble chemical compounds in water. Indirect factors (climate, terrain, water regime, and vegetation, hydrological, and hydrodynamic conditions) produce an environment in which materials can interact with water [2]. The increased nutrients concentration enhanced the eutrophication, and, as a consequence, production of organic matter increases due to increased primary production. This has affected many water bodies, especially since the middle of the 19th century with the development of agricultural intensity and wastewater treatment plants [3]. Despite much research over the last five decades, eutrophication remains a major problem worldwide [4]. Lake ecosystems are particularly susceptible to anthropogenic effects, as they can become a reservoir of water pollutants and a main cause of the deterioration of the ecological status in inland and coastal waters [5,6,7]. In recent years, eutrophication in lakes has been one of the major ecological problems in the world. Most freshwater lakes and wetlands are facing water quality degradation and ecological imbalances due to growing anthropogenic activity, especially in developing countries [8]. Nitrogen and phosphorus compounds are considered to be the key factors in the development of eutrophication in lakes [9,10].

The transformation of nitrogen compounds into sludge and subsequent migration, as well as the reactions between sludge and lake water, are important processes in the biochemical cycle of lake ecosystems [10]. The nitrogen and phosphorus are compounds that promote the primary production in the lakes that can result in the algal blogoms. Phosphorus is usually the limiting nutrient in freshwater, and control of the phosphorus input can improve the water quality [11,12]. It has been found that the “internal” pollution of phosphorus poses a more serious risk in shallow lakes than in deep lakes [13]. The finest particles are usually found in the deepest areas of lake bottoms, where relatively intensive sedimentation occurs due to the production of phytoplankton. Sediment organic matter intensifies biogeochemical nutrient overlays on the sludge surface, such as mineralization, denitrification, or phosphorus relaxation [14]. According to Šaulys [15], phosphorus enters surface waters by being flushed from the soil, rinsed out of rocks, and released as a product of vital activity and the decomposition of aquatic organisms. Bottom sediments are an important element of lake ecosystems because they are involved in internal nutrient cycle processes, and the main role of bottom sediments is to accumulate nutrients [16]. The deposition of phosphorus from sludge to water and its transfer to the trophogenic zone may be more intense than its sorption and deposition [17]. The aim of the studies was to determine the impact of secondary sludge pollution on water. 

## 2. Materials and Methods

Lake sludge studies were carried out, considering the depth of sludge, organic matter content, nitrogen and phosphorus concentrations. 

Five lakes were selected for detailed research. The reason for the choice is the high concentration of N and P in the sludge. A total of 7–10 sludge samples were taken from 5 different Lithuanian lakes: Kiementas, Spėra, Biržulis, Gauštvinis, and Antakmeniai. Figure 1 shows the layout of the lakes. 

Lake Biržulis is located in western Lithuania in the Telšiai district, approximately 5 km northeast of Varniai (Figure 1) in Varniai Regional Park. The length of the lake in the north-south direction is 3.6 km, and the width is 1.3 km. The bottom of Lake Biržulis is covered with sapropelic clay, sludge and clay sapropel in the western and northern bays and sand along the east coast. In the west, the Varna river flows from Lake Lūkštas, and the Nakačia river flows into the south of the lake. Along the Venta tributary, the Virvytė river flows into the north of the lake. There are groundwater springs on the eastern slope of the lake.

Lake Antakmeniai is located in southeastern Lithuania in the Trakai district, approximately 4 km northeast of Aukštadvaris. The length of the lake in the west-southeast direction is 2 km, and the width is up to 0.7 km. The deepest place is 12.5 m.

Gauštvinis Lake is located in the middle of Lithuania in the Kelmė district, approximately 6 km north of Tytuvėnai on the southern edge of the Great Tyruliai peat bog. The length of the lake in the north-south direction is 3.1 km, the width is 0.6 km, and the depth is up to 5 m. The Šimša and Spangupis rivers flow into lake Gauštvinis, and the Dubysa tributary Gryzuva flows into it from the south.

Lake Spėra is located in the Širvintos district of central Lithuania. The maximum lake depth of Lake Spėra is only 2.7 m, and the average depth is 1.85 m. The thickness of the accumulated sludge layer is 6.3 m, and the average layer thickness is 3.0 m.

Lake Kiementas (N55.076553, E25.267448) is located in the southwest of the Molėtai district municipality, east of Giedraičiai. The lake is of a glacial desert origin. The area of the lake is 98.6 ha. The length is 2.4 km, and the maximum width is 0.7 km. The pool area is 23.3 km^2^;. A stream flows into Širvintos through Lake Kiementis and belongs to the Šventoji Basin. The leakage of the lake is 198%. The seabed is covered with muddy sand and deep sludge

### 2.1. Methodology for Sampling and Laboratory Analysis

Sludge and lake water samples were taken from various parts of the lake. The sampling scheme is shown in Figure 2.

A composite sample is made from a 7–10 of discrete samples that have been collected from different locations in the lake and combined into a single sample. This single, composite, sample is representative of the average sludge properties. Sludge and lake water samples were taken under the same natural conditions, namely, sunny days. After the collection of the sludge, all samples were transferred to a laboratory and stored in a refrigerator (5 °C). Sludge samples are taken in glass cylinders during daylight hours according to EN ISO standards:

LST EN 16179: 2012 Sludge, treated bio-waste and soil—Guidance for sample pretreatment [18].

LST EN ISO 5667-13:2011 Water quality—Sampling—Part 13: Guidance on sampling of sludge’s [19]. For sludge collection was used the vacuum sampler. All of the sludge samples, along with the overlaying water, were collected with a glass cylinders (Φ-25 mm, length 5 m), the sludge cores contained 30–40 cm of sediment in a tube, and they were covered by-overlaying water to maintain the sludge structure during sampling and transportation.

LST EN ISO 5667-15:2009 Water quality—Sampling-Part 15: Guidance on preservation and handling of sludge and sediment samples [20].

Water samples are taken according to EN ISO standards: LST EN ISO 5667-14:2016—water quality—Sampling—Part 14 [21].

Sludge samples were filled with fresh surface water. The ratio of sludge to water by weight was 1:10. In this study, 20 g of sludge was collected, and 200 mL of water was added. The prepared samples were stored at a constant temperature of 18 °C under laboratory conditions.

Sludge and water research was conducted at ASU Laboratory in Kėdainiai, with a permit issued by the Minister of the Environment on October 15, 2007, by Order no. D1-522 “Description of Procedure for Issue and Investigation of Pollutant Emissions from Pollutants and Pollutants in Environmental Elements”.

During the study, the total nitrogen (N_t_) and total phosphorus (P_t_) concentrations, electrical conductivity (C) µS/cm, NO_2_–N, NO_3_-N, NH_4_-N and pH values in the water were analyzed. Measurements were made 7, 14, 21 and 28 days after freshwater dilution. The checkpoint was measured on the pickup day.

The following analysis methods were applied: in sludge—total phosphorus (P_t_) studies were performed according to the method LST EN ISO 6878:2004 [22], total nitrogen (N_t_) was tested according to the method LST EN ISO 17184:2014 Soil quality- Determination of carbon and nitrogen by near-infrared spectrometry (NIRS) (ISO 17184:2014) EN ISO 17184:2014 [23], organic matter content was tested according to the method LST EN 13039: Soil improvers and growing media—Determination of organic matter content and ash EN 13039:2011 [24].

In water—total nitrogen (N_t_) was tested according to the method LST EN 13342- 2002 Determination of nitrogen—Determination of bound nitrogen (TNb), following oxidation to nitrogen oxides EN 12260:2003 [25]; total phosphorus (P_t_) studies were performed according to LST EN ISO 6878:2004 [22], electrical conductivity (C) tests were performed according to the method LST EN 27888:1999 [26], pH values—LST EN ISO 10523—2012 Water quality—determination of pH [27], NO_2_–N—LST EN 26777:1999 Water quality—Determination of nitrite—Molecular absorption spectrometric method (ISO 6777:1984) EN 26777:1993 [28], NO_3_-N tests were performed according to the method LST ISO 7890-3:1998 Water quality—Determination of nitrate. Part 3: Spectrometric method using sulfosalicylic acid ISO 7890-3:1988 [29], NH_4_-N tests were performed according to the method LST ISO 7150-1:1998 Water quality. Determination of ammonium. Part 1: Manual spectrometric method ISO 7150-1:1984 [30].

In this study, the lakes were arranged in order from the thinnest to the thickest sludge layer as follows: Biržulis—Antakmenių—Gauštvinis—Spėra—Kiementas.

### 2.2. Statistical Analysis

Correlation and regression were calculated using the computer program STATISTICA 7 [18,19]. The STATISTICA package Nonlinear Estimation was used to determine the correlation coefficients and to define the relationships between the indicators surveyed [31]. The symbol * indicates that the data were reliable within a probability of 95%.

To predict the percentage of total nitrogen, phosphorus and organic matter (%) in the sludge depending on the thickness of the sludge layer and the variable values, IBM SPSS Statistics for Windows, Version 24.0 (IBM Corp., Armonk, NY, USA), was used. Regression formulas were calculated, and scatter diagrams.

Analyses of the selected lake sludge were created.

## 3. Results

Analyses of the 36 lake sludge were carried out by estimating the total phosphorus and total nitrogen concentrations. The results of the study are presented in Figure 3.

Analyses of the selected 5 lake sludge samples were carried out by estimating the total phosphorus and total nitrogen concentrations, the organic matter content and the sludge layer thickness. Since the average lake depth is a very important indicator, it is presented in Table 1 below.

All the analyzed lakes are at risk of increased eutrophication. According to the thickness of the sludge layer, Kiemantas Lake had the largest layer (6.1 m), while Biržulis Lake had the smallest layer (1.9 m). According to the average lake depth, Antakmieniai Lake was the deepest (12.5 m), Biržulis lake was the shallowest (0.9 m). The largest amount of organic matter was in Lake Kiemantas (46.9%), and the smallest was in Gauštvinis Lake (27.6%). The highest concentrations of total phosphorus and total nitrogen were found in Kiemantas and Spėra lakes. Assessing the ecological status of the lakes according to the physico-chemical values according to the Environmental Protection Agency (EPA), Lake Kiemantas showed a very good ecological status (N_t_ 0.90 mg/L and Pt 0.02 mg/L). Based on the total nitrogen concentration (N_t_ 1.0 mg/L) and the total phosphorus concentration (P_t_ 0.081 mg/L), which are average ecological status indicators, Lake Spėra also showed a very good ecological status. According to the state lake monitoring data and based on the total nitrogen concentration (N_t_ 2.15 mg/L), Lake Gauštvinis showed values corresponding to average ecological status indicators. Based on the total phosphorus concentration (P_t_ 0.078 mg/L), Lake Gauštvinis had average state values. Based on the total nitrogen concentration (N_t_ 0.644 mg/L), the values of Antakmenių Lake corresponded to a very good ecological status, while based on the total phosphorus concentration (P_t_ 0.075 mg/L) the values of the lake represented average state values. The total nitrogen concentration (N_t_ 1.25 mg/L) values of Lake Biržulis corresponded to a very good ecological status, while the total phosphorus concentration (P_t_ 0.096 mg/L) values indicated a poor ecological status.

The highest (>10,000 mg/L) concentrations of total nitrogen (N_t_) in sludge were found in Tausalas, Kiementas, Spera, Veisejas, Rėkyva Lake, Judo Kauknorisy, Biržulis, Antakmeniai Lake, Draudeniai Lake, Sablauskiai Pond. Very low (<1000 mg/L) nitrogen concentration in sludge—in Bubliai, Bartkuškis, Janušonys ponds, Luksnėnai, Orija, Skaistė, Kavalis, Alsėdžiai lakes.

The highest (>600 mg/L) concentrations of total phosphorus (P_t_) in sludge were found in the Spera, Kiementas, Draudeniai, Mastis, Antakmeniai, Gauštvinis, Veisiejai, Vaitiekūnai and Stepanioniai ponds. Many of these lakes also contain very high levels of nitrogen and high levels of organic matter. Therefore, five lakes with high concentrations of n and p in sludge were selected for further studies—Birzulis, Antakmenis, Gaustvinis, Spera, and Kiementas.

All the analyzed lakes are at risk of increased eutrophication. Five lakes were selected for detailed research. According to the thickness of the sludge layer, Kiemantas Lake had the largest layer (6.1 m), while Biržulis Lake had the smallest layer (1.9 m). According to the average lake depth, Antakmieniai Lake was the deepest (12.5 m), Biržulis lake was the shallowest (0.9 m). The largest amount of organic matter was in Lake Kiemantas (46.9%), and the smallest was in Gauštvinis Lake (27.6%). The highest concentrations of total phosphorus and total nitrogen were found in Kiemantas and Spėra lakes. Assessing the ecological status of the lakes according to the physico-chemical values according to the Environmental Protection Agency (EPA), Lake Kiemantas showed a very good ecological status (N_t_ 0.90 mg/L and P_t_ 0.02 mg/L). Based on the total nitrogen concentration (N_t_ 1.0 mg/L) and the total phosphorus concentration (P_t_ 0.081 mg/L), which are average ecological status indicators, Lake Spėra also showed a very good ecological status. According to the state lake monitoring data and based on the total nitrogen concentration (N_t_ 2.15 mg/L), Lake Gauštvinis showed values corresponding to average ecological status indicators. Based on the total phosphorus concentration (P_t_ 0.078 mg/L), Lake Gauštvinis had average state values. Based on the total nitrogen concentration (N_t_ 0.644 mg/L), the values of Antakmenių Lake corresponded to a very good ecological status, while based on the total phosphorus concentration (P_t_ 0.075 mg/L) the values of the lake represented average state values. The total nitrogen concentration (N_t_ 1.25 mg/L) values of Lake Biržulis corresponded to a very good ecological status, while the total phosphorus concentration (P_t_ 0.096 mg/L) values indicated a poor ecological status.

After analyzing the data from the collected samples, it can be stated that the amount of biogenic substances in sludge does not always influence the quality of lake water; for secondary pollution to appear from sludge, certain conditions in the water are required.

To predict the amounts of total nitrogen, total phosphorus, and organic matter (%) depending on the thickness of the sludge layer, regression equations were calculated in STATISTICA 8. The scatter charts and regression equations are presented in Figure 4.

After the analysis, the graph shows a linear trend of a positive tendency between the thickness of the sludge layer in the lake and the total amount of nitrogen, phosphorus and organic matter in the sludge of the lake. As the sludge layer increases, the concentrations of nitrogen, phosphorus and organic matter also increase.

To evaluate the intensity of the release of biogenic substances from sludge in freshwater, the N_t_, NO_2_–N, NO_3_-N, and P_t_ concentrations, the specific electrical conductivity µS / cm and the pH values were analyzed. The analyzed data are presented in Figure 5, Figure 6, Figure 7 and Figure 8. Regression equations that can be used to predict water quality values depending on time were calculated. The equations are shown in Table 2, Table 3, Table 4 and Table 5.

The results show that by comparing data from day one to day 28, the NH_4_-N concentration increased from 0.017 mg/L to 0.102 mg/L, the NO_2_-N concentration ranged from 0.039 mg/L to 0.049 mg/L, the NO_3_-N concentration increased from 0.021 mg/L to 0.136 mg/L, and the P_t_ concentration decreased from 0.198 mg/L to 0.006 mg/L. Low concentrations of nitrites, nitrates, and ammonium nitrogen and high concentrations of total nitrogen indicate that the sludge is rich in organic nitrogen. The nitrification process, which takes place in water, is a biological process whereby ammonia is oxidized to nitrite and the latter is oxidized to nitrate.

The results of the study indicate that the concentrations of all the parameters studied increased, with the exception of the total phosphorus concentration. The dynamics of the average phosphorus values are shown in Figure 9.

The average phosphorus values in water decreased from 0.13 mg/L to 0.126 mg/L over 30 days. Therefore, mineral phosphorus (PO_4_-P form) was emitted, and organic phosphorus was “locked” in the sludge.

The same effect can be expected in the lakes that have a low concentration of these elements in the sediment because the lakes were studied don’t have a similar water source and the atmospheric conditions in the region are very similar.

To assess the relationships between the total nitrogen, total phosphorus, nitrates, nitrites, ammonium ions, phosphate, and phosphorus concentrations and the electrical conductivity in water, the STATISTICA 8 software was used to calculate the correlation coefficients based on the data collected, as shown in Table 6.

A strong statistically significant negative correlation between the SEC values and the total phosphorus concentrations indicates that a higher concentration of dissolved salts results in a lower phosphorus concentration in water.

A strong statistically significant positive relationship between the SEC values and the total nitrogen and nitrate concentrations indicates that a higher concentration of dissolved salts results in a greater concentration of total nitrogen and nitrates in water. There is no correlation between the SEC values and ammonium nitrogen with phosphates.

Regression formulas were calculated with the STATISTICA 8 software. Scatter charts and regression equations are presented in Figure 10, Figure 11 and Figure 12.

The graphs in Figure 8 and Figure 9 show a positive correlation between the concentrations of nitrates and total nitrogen in water and the electrical conductivity of the lake water samples and a negative correlation between the concentrations of nitrite and total phosphorus in water and the electrical conductivity of the lake water samples. An increased concentration of dissolved salts resulted in an increased concentration of total nitrates and nitrogen in water. However, if the concentration of the dissolved salts decreased, the total nitrite and phosphorus concentrations in the water decreased. The regression equations can be used to predict changes in the Y (P_t_; N_t_) value when the Y (C µS/cm) value varies. The ammonium nitrogen and phosphate concentrations had no relation to the concentration of dissolved salts.

## 4. Discussion

One of the main biogenic substances that determine the productivity of a surface water body is phosphorus [11]. The study showed that only nitrogen compounds are liberated and can become a source of secondary pollution from sludge in the investigated lakes. Phosphorus accumulates in sludge. These results coincide with the results of a study of the nutrients from Curonian Lagoon and their impact on the ecosystems of the Curonian Lagoon, which were presented on 10-01-2017 [14]. However, Kowalczewska-Madura et al. [32] found that phosphorus release from sludge continues throughout the year in deeper lake areas. Slightly higher values were observed only at the beginning of summer and autumn, and the emitted phosphorus remained in the water layer near the bottom for a long time.

Phosphorus in sediment exists in organic and inorganic forms. Organic phosphorus represents a small fraction of total phosphorus (P_t_), which is generally associated with organisms [33,34,35]. Inorganic phosphorus is typically associated with Fe, Al and Ca compounds [24] and mainly consists of exchangeable phosphorus (Ex-P), ferric phosphate (Fe-P), aluminum phosphate (Al-P), calcium phosphate (Ca-P), and occluded phosphate (O-P) [33]. Fe-P is the most active inorganic phosphorus species, and it has a high release rate. A close correlation between TP release and the Fe-P components in sediment has been reported [35,36,37].

Occluded phosphate (O-P) is adsorbed to a Fe_2_O_3_ or Al_2_O_3_ layer to form a mantle around the phosphoric core, which has a high physical and chemical stability and is not readily available to plants. In addition to phosphorus speciation in sediments, the ecological features of a lake could significantly influence phosphorus release and retention [38].

Studies of the release of biogenic substances from lake sludge into freshwater show differences in the concentrations of biogenic substances released into water. The distribution of biogenic substances in water in the form of fine particles results in a dispersive system. The distributed substance is a dispersed phase (biogenic substance), and the substance in which the dispersed phase is distributed is called the dispersed environment (water). The release of biogenic substances may not be the same in water, and the concentration of dissolved salts is low compared to freshwater, in which dissolved salts have a concentration of approximately 1 mg/L. Dispersed environments are dissimilar, and the distribution of materials is different. 

Studies have shown that an increase in sludge thickness increases the concentrations of nitrogen, phosphorus and organic matter in the sludge. Most data on the internal pollution of water bodies are related to the shallowness of the bodies [17,39,40]; however, little is known about phosphorus release from lakes with greater depths [41], especially those affected by reconstruction [42]. The accumulation of biogenic substances in the bottom sediment depends on the environmental conditions, the morphology of the water body and the hydrology (the fluctuation of the water level and a constant water level). The maximum amount of nutrients accumulates in the deepest part of the water body sediment when the area is adjacent to agricultural property/land [43]. Biogenic substances (nitrogen, phosphorus) accumulate in lake bottoms, and the 10 cm layer of bottom sludge can contain more than 90% of the phosphorus present in the water body itself. Under optimal conditions, nutrients can re-release themselves into the water, which negatively affects the entire lake ecosystem [44,45,46]. Anaerobic processes take place in the thin layer of a few millimeters to a few centimeters of sludge, where organic and inorganic phosphorus forms in orthophosphate in the bottom sediment, and nitrogen compounds turn into ammonia, nitrite or nitrate [44]. Studies have shown that the concentration of dissolved salts in water affects the release of biogenic substances from sludge. The release of total nitrite and phosphorus concentrations from sludge into water are reduced when the values of electrical conductivity are higher. Factors that promote phosphorus release include temperature, pH, concentration of oxygen in the sludge and water layer above the sludge [47], redox potential, type of chemical compounds containing phosphorus, concentrations of Fe, Mn, Al and Ca [38,43,48], bioturbation by microinvertebrates [49] and structure of the bottom sediment [50]. The deposition of phosphorus from sludge to water and its transfer to the trophogenic zone may be more intense than its sorption and deposition [32].

## 5. Conclusions

This research has shown that the thickness of the sludge layer influences the amounts of total nitrogen, phosphorus, and organic matter present in the sludge. As the thickness of the sludge layer increases in a lake, the total concentrations of nitrogen, total phosphorus, and organic matter also increase.

This study on the release of biogenic substances from sludge in freshwater has shown that the concentrations of nitrites (from 0.038 mg/L to 0.049 mg/L), nitrates (from 0.021 mg/L to 0.136 mg/L), phosphates (from 0.317 mg/L to 0.642 mg/L), and total nitrogen (from 7.973 mg/L to 9.06 mg/L) increase, while the concentration of total phosphorus decreases (from 0.198 mg/L to 0.006 mg/L). These results show that organic phosphorus becomes "locked" in sludge, and no secondary water pollution occurs from this source.

Previous studies have shown that the concentration of dissolved salts in water affects the release of biogenic substances from sludge. Due to different dispersion environments, the concentrations of nitrates and total nitrogen increase with the increase in electrical conductivity values. The release of nitrite and total phosphorus concentrations from sludge into water is reduced when higher values of electrical water conductivity are measured.

## Figures and Tables

**Figure 1 ijerph-16-04691-f001:**
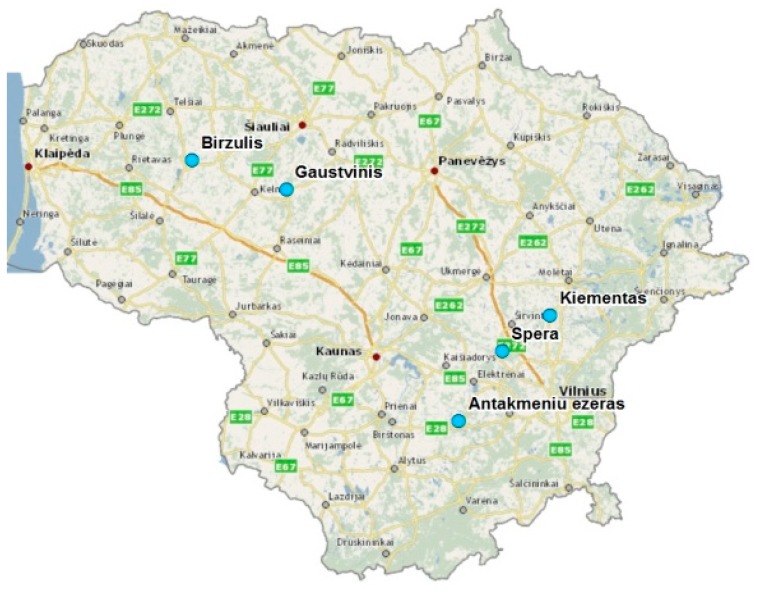
The layout of the investigated lakes in the territory of Lithuania.

**Figure 2 ijerph-16-04691-f002:**
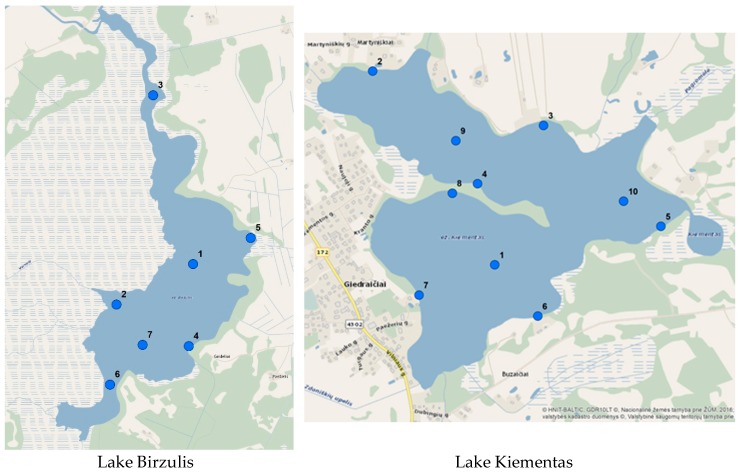
Sampling scheme.

**Figure 3 ijerph-16-04691-f003:**
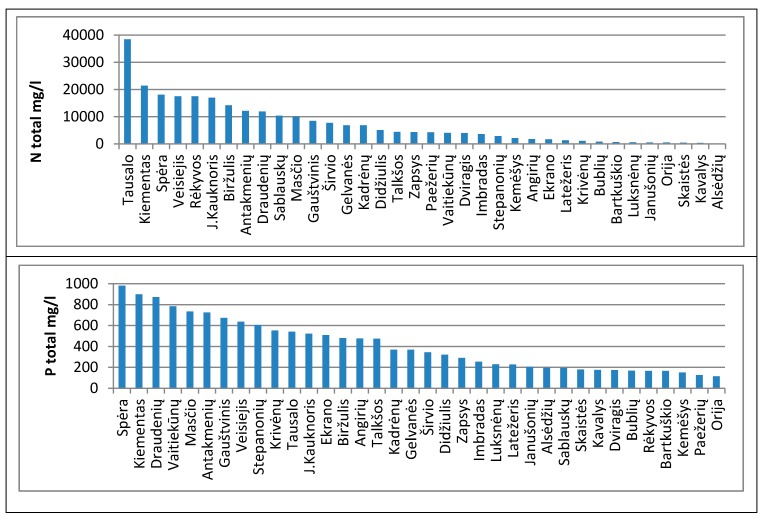
Nitrogen and phosphorus concentration in sludge.

**Figure 4 ijerph-16-04691-f004:**
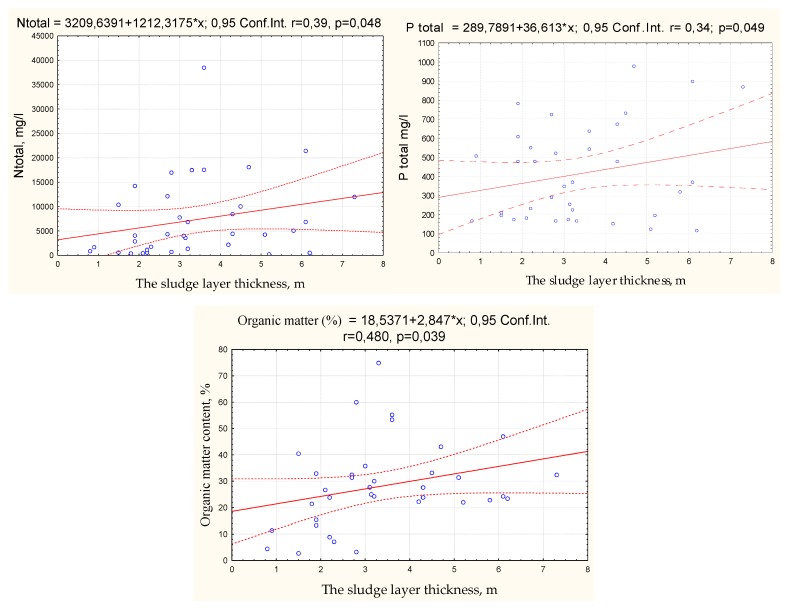
Correlation between N_t_, P_t_ and the amount of organic matter in the sludge to sludge layer thickness.

**Figure 5 ijerph-16-04691-f005:**
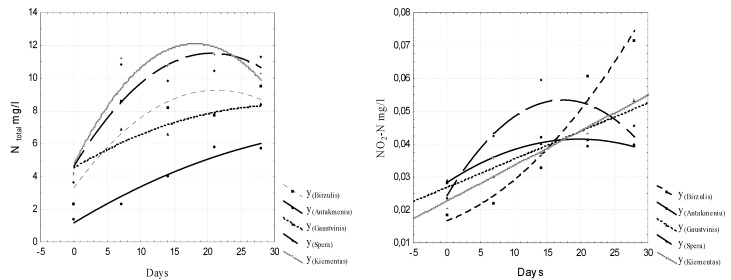
Dynamics of N_t_ and NO_2_-N concentrations in water.

**Figure 6 ijerph-16-04691-f006:**
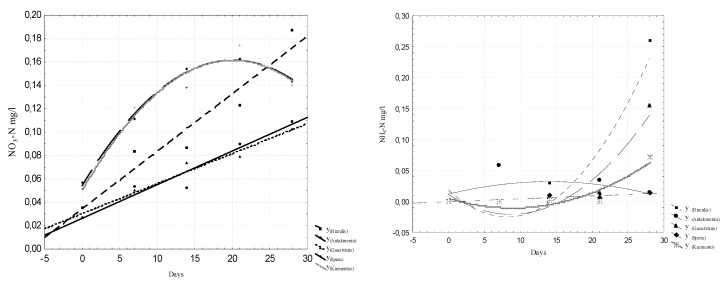
Dynamics of NO_3_-N and NH_4_-N concentrations in water.

**Figure 7 ijerph-16-04691-f007:**
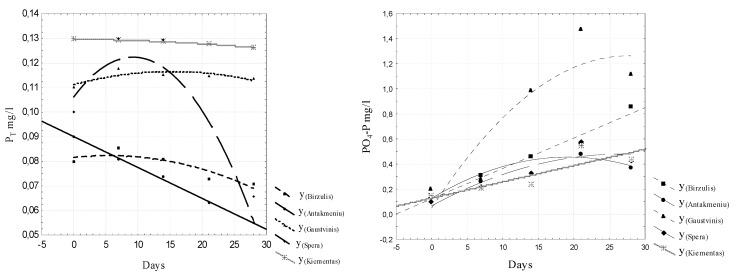
Dynamics of P_t_ and PO_4_-P concentrations in water.

**Figure 8 ijerph-16-04691-f008:**
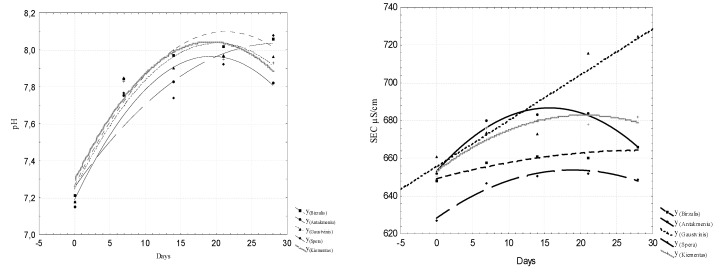
Dynamics of pH value and SEC concentrations in water.

**Figure 9 ijerph-16-04691-f009:**
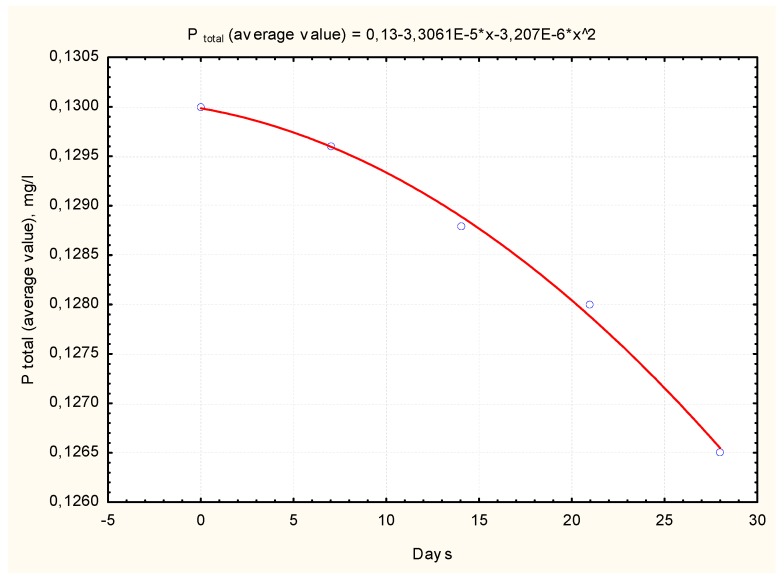
Dynamics of average phosphorus values.

**Figure 10 ijerph-16-04691-f010:**
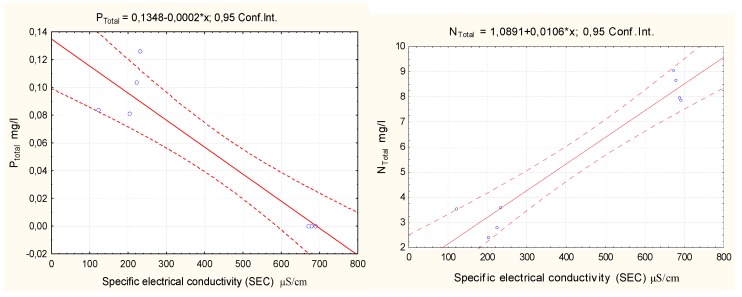
Correlation of N_t_ and P_t_ concentrations with electrical water conductivity.

**Figure 11 ijerph-16-04691-f011:**
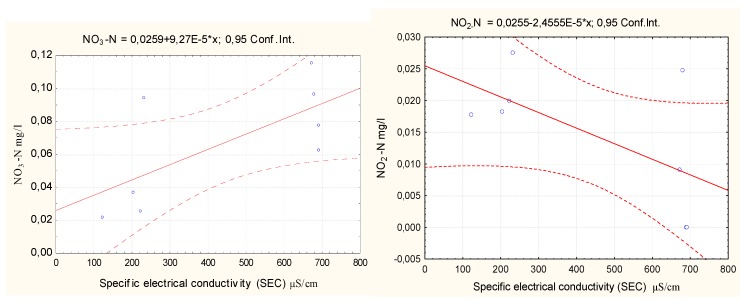
Correlation of NO_3_-N and NO_2_-N concentrations with electrical water conductivity.

**Figure 12 ijerph-16-04691-f012:**
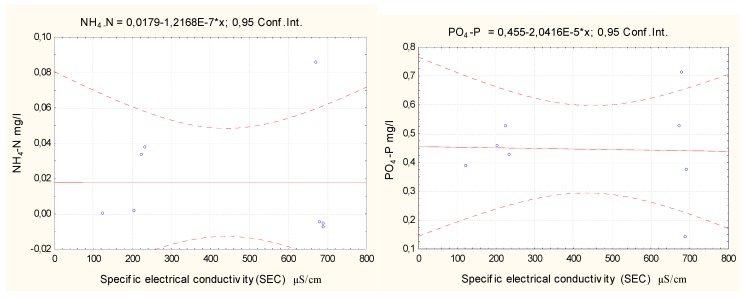
Correlation of NH_4_-N and PO_4_-P concentrations with electrical water conductivity.

**Table 1 ijerph-16-04691-t001:** Depth, sludge layer, N_t_, P_t_ and organic matter content of the sludge samples from the analyzed lakes.

Number	Lake	Average Depth of the Lake, m	Sludge Layer Thickness m	N_T_ (Sludge) mg/L	P_T_ (Sludge) mg/L	Organic Matter Content % (Sludge)
1	Biržulis	0.9	1.9	14,238.4	479.6	32.9
2	Antakmenis	5.8	2.7	12,144.7	725.3	32.4
3	Gauštvinis	5	4.3	8464.5	674	27.6
4	Spėra	1.85	4.7	18,078.0	980.5	43.1
5	Kiementas	4	6.1	21,415.0	898.0	46.9

**Table 2 ijerph-16-04691-t002:** Regression equations for predicting N_t_ and NO_2_-N concentrations.

Number	N Total mg/L	NO_2_-N mg/L
1	Y _Birzulis_ = 3.312 + 0.5577x − 0.0131x^2^, r = 0.863	Y_Birzulis_ = 0.0167 + 0.0008x + 4.6356E − 5x^2^; r = 0.98
2	Y_Antakmeniu_ = 1.1729 + 0.2463x − 0.0026x^2^; r = 0.975	Y_Antakmeniu_ = 0.0283 + 0.0013x − 3.14111E − 5x^2^; r = 0.971
3	Y_Gaustvinis_ = 4.5323 + 0.2396x − 0.0037x^2^; r = 0.944	Y_Gauštvinis_ = 0.0269 + 0.0009x; r = 0.96
4	Y_Spera_ = 4.6234 + 0.6687x − 0.0162x^2^; r = 0.883	Y_Spera_ = 0.0243 + 0.0034x − 9.7668E − 5x^2^; r = 0.885
5	Y_Kiementas_ = 4.8806 + 0.808x − 0.0225x^2^; r = 0.928	Y_Kiementas_ = 0.023 + 0.0011x; r = 0.969 *
0 ≤ x ≤ 2.0		

Note: The symbol * indicates that the data were reliable within a probability of 95%.

**Table 3 ijerph-16-04691-t003:** Regression equations for predicting NO_3_-N and electrical conductivity (C) concentrations.

Number	NO_3_-N mg/L	Electrical Conductivity (C) µS/cm
1	Y_Biržulis_ = 0.0344 + 0.0049x; r = 0.963	Y_Biržulis_ = 649.2857 + 1.0327x − 0.0175x^2^; r = 0.94
2	Y_Antakmeniu_ = 0.0263 + 0.0029x; r = 0.97	Y_Antakmeniu_ = 653.1714 + 4.2939x − 0.137x^2^; r = 0.982 *
3	Y_Gauštvinis_ = 0.0303 + 0.0026x; r = 0.985 *	Y_Gauštvinis_ = 655.8 + 2.4286x; r = 0.938 *
4	Y_Spera_ = 0.0542 + 0.0107x − 0.0003x^2^; r = 0.994 *	Y_Spera_ = 628.4 + 2.7x − 0.0714x^2^; r = 0.98 *
5	Y_Kiementas_ = 0.0515 + 0.0111x − 0.0003x^2^; r = 0.975	Y_Kiementas_ = 653.7429 + 2.8735x − 0.07x^2^; r = 0.942
	0 ≤ x ≤ 2.0	

Note: The symbol * indicates that the data were reliable within a probability of 95%.

**Table 4 ijerph-16-04691-t004:** Regression equations for predicting P_Total_ and PO_4_-P concentrations.

Number	P total mg/L	PO_4_-P mg/L
1	Y _Birzulis_ = 0.0815 + 0.0003x − 2.629E-5 * x^2^; r = 0.909	Y_Biržulis_ = 0.1227 + 0.0242x; r = 0.987
2	Y_Antakmeniu_ = 0.09 − 0.0013x; r = 0.998 *	Y_Antakmeniu_ = 0.1267 + 0.0342x − 0.0009x^2^; r = 0.958
3	Y_Gaustvinis_ = 0.1112 + 0.0007x − 2.2886E − 5 * x^2^; r = 0.781	Y_Gaustvinis_ = 0.0486 + 0.0879x − 0.0016x^2^; r = 0.908
4	Y_Spera_ = 0.1061 + 0.0035x − 0.0002x^2^; r = 0.87	Y_Spera_ = 0.067 + 0.0307x − 0.0006x^2^; r = 0.92
5	Y_Kiementas_ = 0.13 − 3.3061E − 5x − 3.207E − 6Ex^2^; r = 0.998 *	Y_Kiementas_ = 0.1349 + 0.13x; r = 0.82
	0 ≤ x ≤ 2.0	

Note: The symbol * indicates that the data were reliable within a probability of 95%.

**Table 5 ijerph-16-04691-t005:** Regression equations for predicting pH value and NH_4_-N concentrations.

Number	pH	NH_4_-N mg/L
1	Y_Biržulis_ = 7.264 + 0.0783x − 0.0018x^2^; r = 0.974	Y_Biržulis_ = 0.0183 − 0.0106x + 0.0006x^2^, r = 0.92
2	Y_Antakmeniu_ = 7.1877 + 0.0805x − 0.0021x^2^; r = 0.978	Y_Antakmeniu_ = 0.0119 + 0.0029x − 0.0001x^2^, r = 0.39
3	Y_Gauštvinis_ = 7.2446 + 0.0787x − 0.002x^2^; r = 0.958	Y_Gauštvinis_ = 0.0114 − 0.0074x + 0.0004x^2^, r = 0.95
4	Y_Spera_ = 7.2551 + 0.0544x − 0.0009x^2^; r = 0.942	Y_Spera_ = 0.0004 + 0.0005x, r = 0.88
5	Y_Kiementas_ = 7.302 + 0.078x − 0.002x^2^; r = 0.975	Y_Kiementas_ = 0.0061 − 0.0035x + 0.0002x^2^, r = 0.92
	0 ≤ x ≤ 2.0	

**Table 6 ijerph-16-04691-t006:** Correlation matrix between N_t_; NO_2_ -N; NO_3_- N; NH_4_ -NPO_4_-P, P_t_ concentrations and C values.

Parameter	P_T_ mg/L	N_T_ mg/L	NH_4_-N mg/L	NO_2_-N mg/L	NO_3_-N mg/L	PO_4_-P mg/L
SEC µS/cm	r = −0.945 *p* = 0.000	r = 0.966 *p* = 0.000	r = −0.001 *p* = 0.998	r = −0.559 *p* = 0.150	r = 0.787 *p* = 0.049	r = −0.033 *p* = 0.938

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
