# Peer review of "Analysis of Biogenic Secondary Pollution Materials from Sludge in Surface Waters"

_ijerph, 2019, doi:10.3390/ijerph16234691_

Round 1

Reviewer 1 Report

Figure 1 has very low quality, and it cannot be read. In addition, the source from which it was taken must be indicated.

If the data presented in table 1 is the average of all the points sampled in each lake, it is essential to indicate the number of samples and the value of the standard deviation for each of them

The same can be applied to the values indicated between lines 182-200; the number of samples and the standard deviation values found must be included.

Reviewer 2 Report

Dear Authors,

I appreciate the response to the comments given for the first review of this manuscript. Overall I think that there is some improvement in the quality of the paper but some issues still remain. I hope that you realize my only intention is to help you improve your Manuscript with my comments. 

Kind regards.

Reviewer 3 Report

The manuscript has been much improved and is in a nice condition now. I think this manuscript will be acceptable.
